# Dynamic Occupancy Grid Map with Semantic Information Using Deep Learning-Based BEVFusion Method with Camera and LiDAR Fusion

**DOI:** 10.3390/s24092828

**Published:** 2024-04-29

**Authors:** Harin Jang, Taehyun Kim, Kyungjae Ahn, Soo Jeon, Yeonsik Kang

**Affiliations:** 1Graduate School of Automotive Engineering, Kookmin University, 77 Jeongneung-ro, Seongbuk-gu, Seoul 02707, Republic of Korea; tjsdud9536@kookmin.ac.kr (H.J.); martin6773@kookmin.ac.kr (T.K.); kahn@kookmin.ac.kr (K.A.); 2Department of Mechanical and Mechatronics Engineering, University of Waterloo, 200 University Avenue West, Waterloo, ON N2L 3G1, Canada; soojeon@uwaterloo.ca

**Keywords:** occupancy grid map, sensor fusion, autonomous vehicles, semantic grid map, particle filters

## Abstract

In the field of robotics and autonomous driving, dynamic occupancy grid maps (DOGMs) are typically used to represent the position and velocity information of objects. Although three-dimensional light detection and ranging (LiDAR) sensor-based DOGMs have been actively researched, they have limitations, as they cannot classify types of objects. Therefore, in this study, a deep learning-based camera–LiDAR sensor fusion technique is employed as input to DOGMs. Consequently, not only the position and velocity information of objects but also their class information can be updated, expanding the application areas of DOGMs. Moreover, unclassified LiDAR point measurements contribute to the formation of a map of the surrounding environment, improving the reliability of perception by registering objects that were not classified by deep learning. To achieve this, we developed update rules on the basis of the Dempster–Shafer evidence theory, incorporating class information and the uncertainty of objects occupying grid cells. Furthermore, we analyzed the accuracy of the velocity estimation using two update models. One assigns the occupancy probability only to the edges of the oriented bounding box, whereas the other assigns the occupancy probability to the entire area of the box. The performance of the developed perception technique is evaluated using the public nuScenes dataset. The developed DOGM with object class information will help autonomous vehicles to navigate in complex urban driving environments by providing them with rich information, such as the class and velocity of nearby obstacles.

## 1. Introduction

In urban areas, numerous dynamic and static objects such as vehicles, pedestrians, trees, and guard rails coexist. For safe autonomous driving, it is essential to accurately recognize various objects on the road and make safe decisions. Therefore, it is necessary to integrate environmental perception results from multiple sensors to effectively represent the road environment. One prominent technique among such representation methods is the grid map. The grid map collects sensor information to depict the surrounding environment from a bird’s eye view (BEV) perspective in a grid format. Therefore, it is advantageous for representing the position of an object in the surrounding urban environment. Conventional static occupancy grid maps provide only the position information of the objects in each grid. Thus, it is impossible to represent the dynamic state of the objects. To overcome this limitation, dynamic occupancy grid maps (DOGMs) have been developed, which facilitate the dynamic states of objects occupying a grid cell [1]. This capability provides a more accurate understanding of the situation by distinguishing between dynamic and static states for each object. In recent years, numerous studies have explored methods for generating dynamic grid maps using various sensors.

Figure 1a presents the results of generating a DOGM using light detection and ranging (LiDAR). Here, a challenge arises in misclassifying static objects, such as walls, as dynamic objects due to occlusion by nearby objects. This misclassification can potentially lead to erroneous path prediction and decision making during autonomous driving. Thus, the objective of this study is to enhance object detection and recognition capabilities by leveraging camera–LiDAR sensor fusion, as depicted in Figure 1b. In particular, a more accurate classification of dynamic and static states is possible by considering the semantic information of objects. For more accurate velocity estimation, we have developed two different update models to incorporate oriented bounding boxes (OBBs), the outcomes of object detection from deep learning techniques, on the grid map. Virtual points have been generated to represent the OBBs on the grid, and occupancy probabilities and label information have been assigned to the grid cells, including the virtual points. This approach is a contrast to conventional LiDAR-based DOGM generation [1]. Yet, the use of virtual points for OBBs, as shown in Figure 1b, allows for a more accurate representation of the shape of a vehicle.

The schematic of this study is shown in Figure 2, which is divided into two parts: one using unclassified LiDAR points to build a conventional static occupancy grid map, and the other using camera and LiDAR measurements for deep learning-based object detection to build a dynamic occupancy grid map. The results from the object detection are classified into four movable labels—cars, bikes, pedestrians, and others—in a form of oriented bounding box.

Our method assigns occupancy probabilities to these dynamic objects through a particle filter. Meanwhile, LiDAR points that are not classified as movable labels are utilized to construct a static occupancy grid map. As a result, we can achieve robust environmental perception by representing not only dynamic states such as speed and angle, but also static environmental information in a BEV format. Furthermore, this study compares two sensor models for extracting more accurate velocity information from OBBs. The two models differ in how occupancy probabilities are assigned: one uniformly assigns probabilities within the bounding box, whereas the other assigns probabilities only along the perimeter of the box. By comparing the speed estimation results of these two models with different occupancy probability assignment methods, we propose a sensor model that allows for more accurate speed estimation.

Our main contributions are summarized as follows:The robustness of environmental perception is enhanced by developing a DOGM that incorporates deep learning-based object recognition results through the BEVFusion camera–LiDAR sensor fusion method.More accurate classification of the dynamic and static states of objects is realized by leveraging semantic information with potential dynamic states as input for the DOGM, thereby estimating not only the position and class of each object, but also inferring its velocity.Performance of the estimation of the speed and direction of dynamic objects is enhanced by using the edge bounding box update model (EBBUM) for potential dynamic objects classified using camera–LiDAR sensor fusion.

## 2. Related Work

### 2.1. Sensor Fusion of LiDAR and Camera

Various sensors are commonly used in autonomous driving, such as cameras, LiDAR, and radar. Cameras have the advantage of extracting a wealth of information about the surrounding environment, leading to extensive research on object detection using cameras. However, a drawback of cameras is their inability to perceive depth from a single camera image.

In contrast, a LiDAR sensor offers the advantage of providing precise distance measurements in a 3D format.

Recently, numerous studies on deep learning-based object detection methods using sensor fusion have been conducted. Methods for deep learning 3D object detection using cameras and LiDAR can be categorized into early, middle, and late fusion. Early fusion integrates raw data from different sensors. An example is the fusion of raw data from a camera–LiDAR sensor, which creates RGB-D information by combining features at the raw data level. This approach is used to generate feature maps for object detection [2]. Middle fusion involves combining features or information extracted at an intermediate step. Although it integrates raw data from each sensor, similar to early fusion, the distinction lies in the use of separate neural networks for fusion. A study on a methodology that operates the LiDAR data voxelization pipeline and the camera pipeline in parallel has been conducted. Subsequently, features from each sensor are fused, considering the disparity in their importance, and are used to refine the final region proposal [3].

Late fusion involves an independent information process, followed by a combination of features obtained from each sensor. Some studies have enhanced the detection performance by leveraging geometric and semantic consistencies prior to the non-maximum suppression stage of the detector [4]. As demonstrated by various research efforts, sensor fusion techniques efficiently combine features from different sensors, addressing weaknesses while retaining their strengths, which makes them advantageous for object recognition. Thus, in this study, we perform an object recognition technique using camera–LiDAR sensor fusion to enhance object detection performance.

### 2.2. Dynamic Occupancy Grid Maps (DOGMs)

A DOGM is an environment perception technique used in robotics and autonomous driving applications. A DOGM has the advantage of not only detecting static obstacles, but also identifying the position and velocity of dynamic objects, allowing safe navigation in dynamic environments. Several studies are currently being conducted on DOGMs, with an emphasis on estimating the states of dynamic and static objects and velocity [5]. Additionally, research has been conducted on the fusion of multiple radar sensors, instead of a single laser. This research also includes the detection of dynamic objects using grid-based object tracking and mapping methods, in conjunction with clustering algorithms [6]. Research is not limited to creating DOGMs using only radar sensors; there are ongoing studies on generating DOGMs by employing various sensors. One such study uses inverse perspective mapping images and LiDAR to create DOGMs [7]. Efforts are also underway to perform object-level tracking using DOGMs in dynamic environments [8,9]. Furthermore, research has been conducted on applying deep neural networks, including recurrent neural networks (RNNs) and deep convolutional neural networks, to DOGMs for estimating the states of dynamic objects [10,11].

In general, it is computationally expensive to maintain a large number of particles to represent a complex map. However, as shown in Figure 3, a substantial portion of the surviving particles are drawn towards static objects, such as bushes, trees, and buildings, leading to fewer surviving particles being allocated to dynamic objects. This phenomenon makes it challenging to maintain stable velocity estimation performance for moving objects, resulting in the misclassification of static objects as dynamic and a decrease in the accuracy of object velocity estimation.

The class information of objects is valuable for understanding urban environments, particularly in areas with various types of static and dynamic objects. In this context, research has been conducted on integrating maps with semantic information using LiDAR, specifically employing the PointPillar approach for semantic labeling. This study combines spatiotemporal and conditional variational deep learning methods to detect vehicles [12]. Another study combines DOGMs and object-level tracking using LiDAR. The tracked objects are labeled with semantic information considering temporal changes [13]. Furthermore, research has been conducted on the development of semantic DOGMs using deep learning techniques. One study employed a multitask RNN with LiDAR point clouds as input to predict occupancies, velocity estimates, semantic information, and drivable areas on a grid map [14]. Studies have also used stereo cameras, not only LiDAR. One research effort involved extracting semantic and distance information by generating depth maps and using image segmentation to estimate the occupancy grid map [15,16]. Another study used monocular RGB images to process semantic segmentation, employing Bayesian filters to fuse occupied grids and estimate semantic grids [17]. However, using only a camera to create an occupancy grid map can lead to inaccurate distance information, and relying solely on LiDAR may lead to low object detection performance, making it challenging to discern objects. Therefore, the objective of this study is to enhance object detection performance by using both a camera and LiDAR as inputs, enabling a more accurate understanding of the current situation of the driving vehicle.

## 3. Methods

### 3.1. 3D Object Detection

Sensor fusion for object recognition has been extensively researched to enhance object detection capability by compensating for the weakness of each sensor through a combination of multiple sensors [18,19,20]. BEVFusion is a method developed for 3D object detection using only a camera or a combination of a camera and LiDAR. This method generates feature maps for both the camera and LiDAR and fuses them for object recognition, thereby maintaining the advantages of both the camera and LiDAR [21]. BEVFusion represents the features of cameras and LiDAR sensors in the BEV format, providing a framework for sensor fusion. Therefore, the distance and semantic information are preserved to enhance 3D recognition performance. In this study, BEVFusion utilizes a camera and LiDAR fusion to generate OBBs, which serve as input for DOGMs. This enables the estimation of not only the position and class, but also the velocity information of objects.

Additionally, an environmental perception map was created using LiDAR points that were not classified as movable labels. The developed grid map classified dynamic and static objects in urban environments while extracting semantic information about dynamic objects, such as vehicles, motorcycles, bicycles, and pedestrians. To find the grids belonging to the OBB generated by 3D object detection, virtual points were created within the bounding box using the coordinates of the four vertices, as illustrated in Figure 4. The virtual points ($x_{box}$) consist of the x and y coordinates ($p_{x},\;p_{y}$), the centroid x and y coordinates of the detected object ($P_{c,}$), the yaw angle (θ), and the label information (l). Similar to how LiDAR point clouds occupy grids, the virtual points within the bounding box assign occupancy probabilities to the DOGM.
(1)$$x_{box}=\left[\;p_{x},\;p_{y},\;v_{x},\;v_{y},\theta ,\;l,P_{c}\;\right].$$

Using the OBB obtained from deep learning as input to the DOGM, the measurement update model influences the survival of the predicted particles, leading to performance variations in the velocity estimation. Therefore, in this study, we employed two different update models for grid updates: the solid bounding box update model (SBBUM) and the EBBUM.

Figure 5a shows the SBBUM, which assigns a uniform occupancy probability to the grids corresponding to the box area; and Figure 5b shows the EBBUM, which assigns occupancy probability only to a specific region corresponding to the box border. The thickness (T) of the border was set to 0.4 m, considering a sufficient number of particles to represent the shape of the vehicle and the grid size of the grid map.

### 3.2. Sensor Fusion DOGM

In many studies that adopt DOGMs, the occupancy information of grid cells is typically measured using radar or LiDAR sensors. In this study, we developed an algorithm for generating virtual points based on the OBB estimation results for grid cells.

In particular, we reduced the computational load by assigning particles only to grid cells occupied by objects with potentially movable classes, such as vehicles, motorcycles, bicycles, and pedestrians. This reduction in computational load is particularly significant when the number of moving objects is small.

The generated grid map contains Occupancy (O), Free (F), and Label (L) information within each grid cell, as shown in Figure 6, where the Label (L) consists of car ${(l}_{c})$, bike ${(l}_{b})$, pedestrian ${(l}_{p})$, and other ${(l}_{o})$. The DOGM with semantic information is updated by the following steps.

Initialization

At time *t*, the state of the grid map is updated using particle filters. When there are $\nu^{c}$ particles associated with the occupied grid cell c, the particles at time t include information such as their coordinates ${\left\{x_{t}^{\left(i,c\right)}\right\}}_{i=1}^{\nu_{t}^{c}}$, weights ${\left\{{\;w}_{t}^{\left(i,c\right)}\right\}}_{i=1}^{\nu_{t}^{c}},$ and label weights ${\left\{l_{t}^{\left(i,c\right)}\right\}}_{i=1}^{\nu_{t}^{c}}$. These are used to assign Occupancy mass ($m_{t}^{c}\left(O\right))$ and Label mass ($m_{t}^{c}\left(L\right))$ using (2).
(2)$$m_{t}^{c}\left(O\right)=\;\sum_{i=1}^{v_{t}^{c}}w_{t}^{\left(i,c\right)}.\;m_{t}^{c}\left(L\right)=\;\sum_{i=1}^{v_{t}^{c}}l_{t}^{\left(i,c\right)}.$$

Prediction

At time t, the particle configurations within the measured grid are predicted for time *t* + 1 using the constant velocity prediction model. “+” denotes the prediction step from time *t* to *t* + 1. The weights of the predicted particles ${\left\{w_{+}^{\left(i,c\right)}\right\}}_{i=1}^{\nu_{+}^{c}}$ estimate whether the grid at time *t* + 1 is occupied or unoccupied. The predicted label mass is computed using the label weights of the predicted particles ${\left\{l_{+}^{\left(i,c\right)}\right\}}_{i=1}^{\nu_{+}^{c}}$. Equation (3) calculates the mass of the predicted occupied grid ($m_{+}^{c}\left(O\right))$, and the label mass of the predicted grid ($m_{+}^{c}\left(L\right)).$ The free mass is determined by taking the smaller of two masses: the predicted occupied mass and the free mass, adjusted by applying the discount factor α(t) over time t. In Equation (4), α(t) denotes the discount factor representing the decrease in confidence over time intervals.

In this case, the sum of the particle and label weights for each grid is constrained to be 1. If the sum of the predicted occupied weight ($m_{+}^{c}\left(O\right)$) or the sum of the label weights ($m_{+}^{c}\left(L\right)$) exceeds 1, normalization is performed. Labels (L) for $m_{+}^{c}\left(L\right)$ in this study consist of car ($l_{c}$), bike ($l_{b}$), person ($l_{p}$), and other ($l_{o}$).
(3)$$m_{+}^{c}\left(O\right)=\;\sum_{i=1}^{\nu_{+}^{c}}w_{+}^{\left(i,c\right)}.\;m_{+}^{c}\left(L\right)=\;\sum_{i=1}^{\nu_{+}^{c}}l_{+}^{\left(i,c\right)}$$
(4)$$m_{+}^{c}\left(F\right)=\min\left[\alpha \left(t\right)m_{t}^{c}\left(F\right),\;1-m_{+}^{c}\left(O\right)\right],\;\;\alpha \left(t\right)\;\in \left[0,1\right].$$

Update

During the update step, the combination of the measurement mass $m_{\varrho }$ and predicted mass $m_{+}$ is performed using the Dempster–Shafer evidence theory using (5), where the event defined by A and B corresponds to the hypothesis of occupancy, freeness, and uncertainty. K represents the sum of the masses of an empty set defined by mutually exclusive events between Occupancy and Free.
(5)$$m_{t+1}\left(O\right)=\left(m_{\varrho }⊕\;m_{+}\right)\left(O\right)\;=\;\frac{1}{1-K}\sum_{A\cap B=O}m_{\varrho }(A)\;m_{+}\left(B\right).$$

The label estimation step defined by (6) also employs the Dempster–Shafer evidence theory to combine the measurement and predicted mass of label information.
(6)$$m_{t+1}\left(L\right)=\left(m_{\varrho }⊕m_{+}\right)\left(L\right)\;=\;\frac{1}{1-K}\sum_{l_{1}\cap l_{2}=L}m_{e}\left(l_{1}\right){\;\;m}_{+}\left(l_{2}\right).$$

In (6), $l_{1}$ and $l_{2}$ represent the event defined by the hypothesis of the four labels. Here, K represents the sum of the masses of empty labels defined by mutually exclusive events between labels. 

At time t + 1, occupancy mass ($m_{t+1}^{c}(O)$) comprises persistent cells ${(m}_{p,t+1}^{c})$ and newborn cells ${(m}_{b,t+1}^{c})$. Persistent cells represent grids that existed in the map at the previous and subsequent steps. Newborn cells denote grids occupied by an object in the t + 1 step that were not occupied in the previous time step. $p_{B}$ denotes a birth probability of the newborn cells. The mass of newborn cells is distributed between the predicted and measured masses through the probability of $p_{B}$.

The occupancy mass of the newborn cell ($m_{b,t+1}^{c})$ is calculated using (7). Similarly, the label mass of the newborn cell ($m_{b,l,t+1}^{c})$ is determined by (8).
(7)$$m_{b,t+1}^{c}=\;\frac{m_{t+1}^{c}\left(O\right)\cdot p_{B}[1-m_{+}^{c}\left(O\right)]\;}{m_{+}^{c}\left(O\right)+p_{B}[1-m_{+}^{c}\left(O\right)]}.$$
(8)$$m_{b,l,t+1}^{c}=\;\frac{m_{t+1}^{c}\left(L\right)\cdot p_{B}[1-m_{+}^{c}\left(L\right)]\;}{m_{+}^{c}\left(L\right)+p_{B}[1-m_{+}^{c}\left(L\right)]}.$$

Equation (9) represents the occupancy mass of the persistent cell ($m_{p,t+1}^{c}$), which is calculated by subtracting the mass of the newborn cells from the occupancy mass of the grid.
(9)$$m_{p,t+1}^{c}=\;\;m_{t+1}^{c}\left(O\right)-m_{b,t+1}^{c}.$$

Similarly, (10) expresses the label mass of the persistent cell ($m_{p,l,t+1}^{c}$), which is calculated by subtracting the label mass of the newborn cell from the label mass of the grid.
(10)$$m_{p,l,t+1}^{c}=\;\;m_{t+1}^{c}\left(L\right)-m_{b,\;l,t+1}^{c}.$$

The mass of the persistent cell in a grid is composed of occupancy mass ($m_{p,t+1\;}^{c})$ and label mass ($m_{p,l,t+1}^{c})$. They are calculated by the weights for each particle ($w_{p,t+1}$) and label ($l_{p,t+1})$ normalized by the total number of persistent particles ($\nu_{p}^{c})$ using (11).
(11)$$w_{p,t+1}=\frac{m_{p,t+1\;}^{c}}{\nu_{p}^{c}}.\;l_{p,t+1}=\frac{m_{p,l,t+1\;}^{c}}{\nu_{p}^{c}}.$$

Similarly, the weights associated with the occupancy mass ($m_{b,t+1\;}^{c})$ and the label mass ($m_{b,l,t+1\;}^{c})$ for the newborn grids are calculated using (12).
(12)$$w_{b,t+1}=\frac{m_{b,t+1\;}^{c}}{\nu_{b}^{c}}.\;l_{b,t+1}=\frac{m_{b,l,t+1\;}^{c}}{\nu_{b}^{c}}.$$

These weights are normalized by the total number of newborn particles ($\nu_{b}^{c})$.

Resampling

Finally, resampling is performed to generate new particles, reducing the uncertainty of the particle filter and leading to more accurate estimations. In each grid, the weights of persistent and newborn particles are combined, and the particles are resampled on the basis of their weights. New particles are assigned uniform weights. In this step, particles with lower weights are eliminated, and the overall number of particles is maintained by resampling with higher weights.

## 4. Experiments

### 4.1. Dataset

The main sensors in the nuScenes dataset include one 32-channel LiDAR, six cameras, and one radar. Table 1 explains the specifications of the sensors in the nuScens data set. The dataset contains measurements from the LiDAR, updated at 20 Hz, and six cameras with a resolution of 1600 × 1200 pixels, captured at 12 Hz [22]. The nuScenes dataset comprises scenarios acquired in complex urban areas in Boston and Singapore. It comprises 1000 scenes, each lasting approximately 20 s, and includes labeling for 23 classes such as car, bus, bicycle, motorcycle, pedestrian, and barrier. For validation and analysis, two scenarios were selected: one with a dense forest and pond, and another with multiple objects, including bicycles.

### 4.2. Evaluation Process

In this study, the sensor fusion DOGM provides information about the dynamic and static states and velocity of each object. The algorithm developed in this study is validated by comparing the estimated velocity values from the sensor fusion dynamic grid map with the ground truth velocities annotated in the nuScenes dataset. Furthermore, OBBs contain angle information. Precise angle estimation is important to predict the path of objects. By using the OBBs as inputs for the sensor fusion DOGM, not only the velocity, but also the direction of the object is estimated. Consequently, this study analyzes the direction error of the estimated box using the sensor fusion dynamic grid map, and compares it with the ground truth obtained from the OBB provided in the nuScenes dataset.

## 5. Results and Discussion

### 5.1. Scenario Involving Diverse Objects Such as Vehicles, Bicycles, and Pedestrians (Scene No. 98)

In this scenario, various types of objects such as cars, pedestrians, and bicycles are commonly found in urban areas. Furthermore, there are both dynamic objects, such as cars and pedestrians, and static objects, such as traffic cones. The specific situation involves the ego vehicle traveling in a straight line, followed by a bicycle from behind. In this study, various objects were detected using a 3D object detection method called BEVFusion. Figure 7 shows the results validated using the nuScenes dataset with BEVFusion. The results demonstrate detection performance not only for large vehicles, but also for smaller objects such as bicycles and traffic cones. Figure 8 presents the BEVFusion results in the BEV format, showing stable detection even for objects located beyond 50 m. This demonstrates robust detection performance in complex urban environments with multiple vehicles. To further analyze the performance, we compared two cases based on the occupancy state of the bounding boxes using BEVFusion’s 3D object bounding box as input for the sensor fusion DOGM.

Figure 9 shows the results of the sensor fusion DOGM using the position and class information from the OBB as input. Figure 9a,c show the results when the SBBUM is used as the OBB input, and Figure 9b,d show the results when the EBBUM is used as the OBB input. The vehicles in Figure 9a,c occupy grids in a solid square shape, whereas those in Figure 9b,d occupy grids on the edge of the square shape. In both cases, stable dynamic and static classifications were observed even in scenarios with multiple vehicles and bicycles. For small objects, such as pedestrians, both cases accurately performed dynamic and static classification. The enlarged circular plots in Figure 9a,c show the velocity estimation of the bike behind the ego vehicle and the ground truth. Because the bike was making a left turn at the moment, the SBBUM had slower convergence to the ground truth than the EBBUM in estimating the orientation of the vehicle.

Figure 10 compares the estimated speed results of a bicycle following the ego vehicle in nuScenes scenario 98 with the ground truth speeds provided by the nuScenes dataset. The SBBUM exhibited a mean absolute speed error of 0.581 m/s, whereas the EBBUM exhibited an error of 0.599 m/s. Both models estimated speeds accurately compared with the ground truth values from the nuScenes dataset. Figure 11 compares the angle estimation of a bicycle following the ego vehicle with the ground truth. The angle estimation has a mean absolute error of 0.227 radians for the SUBBM and 0.135 radians for the EBBUM. The proposed EBBUM has a significantly lower angle estimation error than the SUBBM.

### 5.2. Scenarios Involving Many Bushes, Trees, Etc. (Scene No. 272)

In this scenario, the environment consists of static objects, such as trees and roadside structures. The ego vehicle follows a preceding vehicle that travels straight and then makes a left turn. Furthermore, another vehicle is behind the ego vehicle, and there are stationary motorcycles to the right of the ego vehicle. Figure 12 shows the results of using BEVFusion for object detection. The figure confirms the accurate recognition performance of the employed method for detecting various objects, such as motorcycles and cars, marked over the camera images. Figure 13 shows the BEVFusion results in the BEV format. Figure 13a,c represent the ground truth from the nuScenes dataset, and Figure 13b,d depict the outcomes after applying BEVFusion. The results show stable and reliable recognition of diverse objects.

Figure 14 shows a scenario in which other vehicles are traveling in front of and behind the ego vehicle. The OBBs generated by BEVFusion were used to create sensor fusion DOGMs. Figure 14a,c show the results when the SBBUM is used for the OBB input, whereas Figure 14b,d show the results when the EBBUM is used for the OBB input. In both cases, accurate dynamic recognition of objects in front of and behind the ego vehicle was confirmed. The recognition of a stationary motorcycle along the roadside verifies the robust performance of the DOGMs. However, Figure 14a,b reveal that when the EBBUM is used, the accuracy of the direction of the velocity vector is more precise.

In Figure 15a compares predicted and persistent particles when the SBBUM is used, and Figure 15b shows the particles when the EBBUM is used. In the SBBUM’s case, the predicted particles within the object bounding box maintain their original direction even when the object rotates. This can lead to delayed convergence to the ground truth, resulting in many estimation errors in the orientation when the object is making a turn. To improve convergence, the proposed EBBUM in Figure 15b assigns probability only along the edge of the bounding box to help the survival of the predicted particles aligned with the direction of the OBB measurement. Consequently, particles predicted in the direction of the original motion are eliminated because the EBBUM assigns a low probability mass inside the bounding box. However, particles predicted in the direction of object rotation are more likely to survive, resulting in better accuracy of the velocity estimations of rotating objects. Therefore, it is expected that the EBBUM will provide a more accurate direction estimation of an object than the SBBUM, especially when the object is making turns.

Figure 16 compares the estimated velocity results of the vehicle following the ego vehicle in nuScenes scenario 272 with the ground truth velocity. The graph shows the velocity estimation results for both the SBBUM and EBBUM cases, along with the ground truth velocity provided by nuScenes. Compared with the ground truth velocity from nuScenes, case (a) had a mean absolute speed estimation error of 0.4945 m/s and case (b) had a mean absolute speed estimation error of 0.5037 m/s. Both cases initially exhibited differences in speed estimation because of the convergence times of particles; however, over time, the velocity errors between the ground truth and sensor fusion DOGMs decreased.

Figure 17 compares the direction estimations between the SBBUM and EBBUM. The SBBUM exhibited small errors in estimating the direction at the beginning. However, after 4.5 s, the vehicle following the ego vehicle started to rotate, and the velocity estimation results of the EBBUM converged faster to the ground truth direction than those of the SBBUM. In this result, the SBBUM had a mean absolute error of 0.153 radians, whereas the EBBUM reduced the error by 37%, resulting in 0.097 radians. This reduction in errors is attributed to the faster convergence of the direction estimates to the ground truth during the turn.

## 6. Conclusions

In this study, we propose a robust perception method for urban environments, based on state-of-the-art DOGM technology combined with semantic information obtained from the BEVFusion results of camera–LiDAR sensor fusion. Using the Dempster–Shafer theory of evidence, a method of updating occupancy and class information of a grid map has been developed, as well as velocity information, which is important for an autonomous vehicle with uncertain sensor information to navigate in an urban environment, where different types of dynamic and static objects coexist.

OBBs are outputs commonly used as fusion results of LiDAR sensors and cameras. To employ the OBB as measurement input to a DOGM, virtual points are assigned to the grid cell occupying the OBB. Furthermore, two different update models are employed for a sensor fusion-based DOGM using OBBs: one assigns the occupancy probability to the entire grid, including the OBB (SBBUM), and the other assigns only the occupancy probability on the edges of the OBB (EBBUM). The speed and angle estimates are employed as evaluation metrics to compare the mean absolute errors from the ground truth provided by the public nuScenes dataset. In the case of the SBBUM, the speed errors for vehicles and bicycles were 0.4945 and 0.581 m/s, respectively. For the EBBUM, speed estimation errors of 0.5037 and 0.599 m/s were obtained for vehicles and bicycles, respectively. Both methods exhibited approximately similar performances in speed estimation. However, there was a significant improvement in estimating the directions of dynamic objects, especially during the scenario in which the moving directions of objects changed. During this scenario, the mean absolute heading angle error for the SBBUM was 0.153 radians, whereas that of the EBBUM was 0.097 radians. The proposed EBBUM showed a better angle estimation performance because the particles predicted in the direction of object rotation are more likely to survive.

In conclusion, the proposed method can be used to incorporate different results from deep learning studies to construct a map of the surrounding environments with uncertainty. Because the constructed grid map is from a BEV perspective, it can also be used for motion prediction of nearby dynamic objects or path planning of the ego vehicle.

In a future study, we plan to employ the outputs from different deep learning-based classification methods, such as semantic segmentation, to construct DOGMs. This extension will allow the DOGM to contain richer information of nearby environments, both static and dynamic.

## Figures and Tables

**Figure 1 sensors-24-02828-f001:**
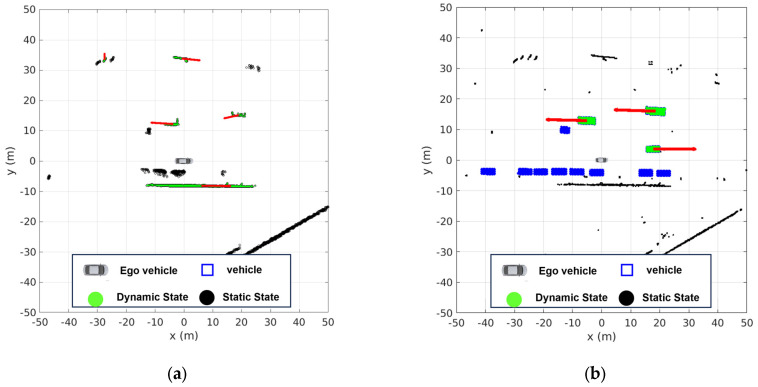
Example of DOGM using (**a**) only LiDAR measurements and (**b**) LiDAR and semantic information.

**Figure 2 sensors-24-02828-f002:**
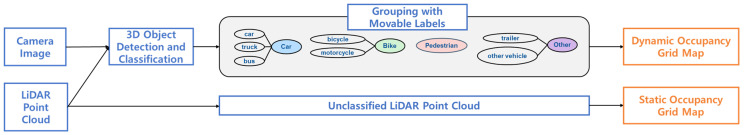
Camera–LiDAR sensor fusion DOGM framework overview.

**Figure 3 sensors-24-02828-f003:**
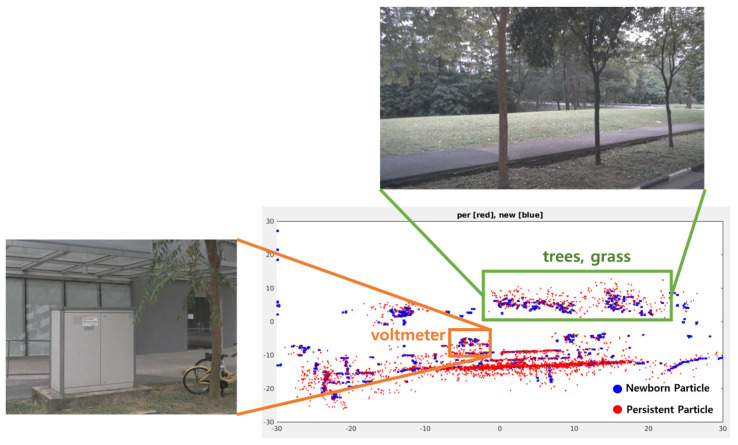
Persistent and newborn particle distribution of LiDAR-based DOGM.

**Figure 4 sensors-24-02828-f004:**
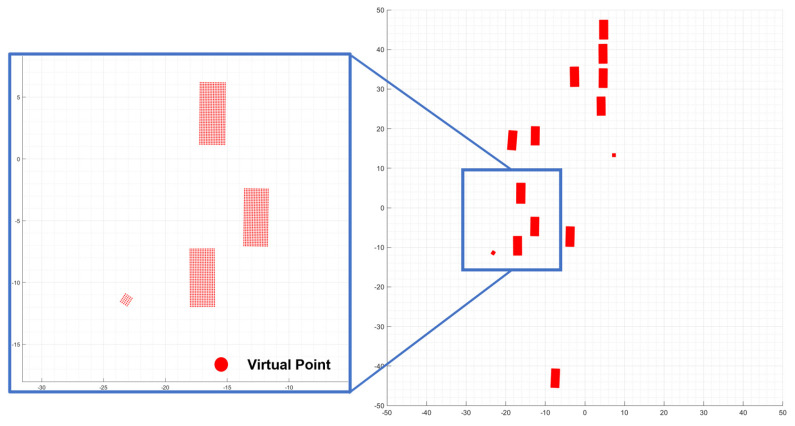
Virtual point assignment of OBBs to DOGMs.

**Figure 5 sensors-24-02828-f005:**
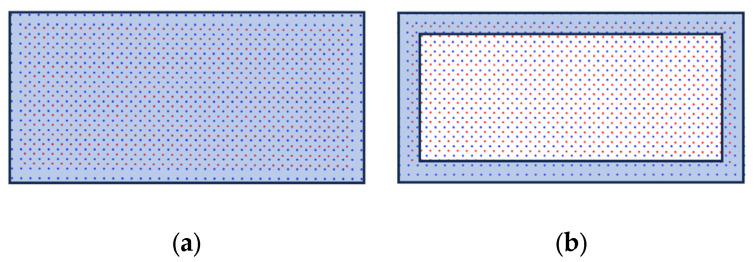
Virtual point assignment using (**a**) SBBUM and (**b**) EBBUM.

**Figure 6 sensors-24-02828-f006:**
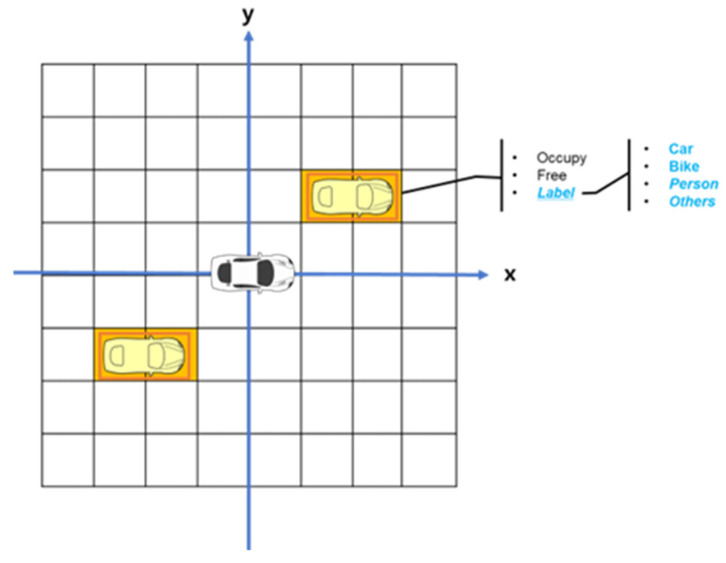
Configuration of semantic occupancy information in each cell.

**Figure 7 sensors-24-02828-f007:**
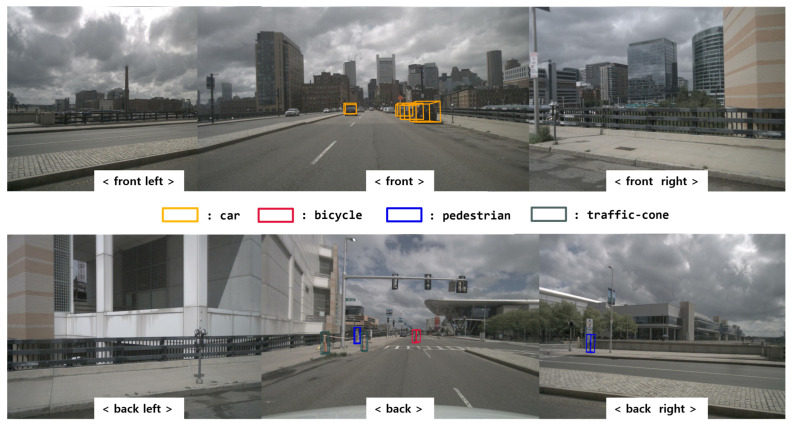
BEVFusion result with the camera image (scene no. 98).

**Figure 8 sensors-24-02828-f008:**
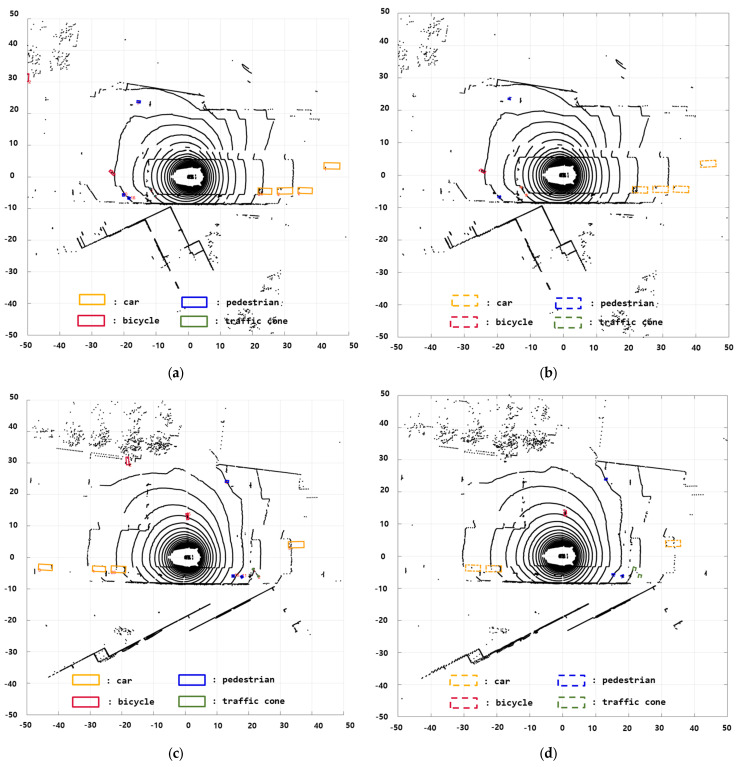
BEVFusion results with LiDAR measurements: (**a**,**c**) nuScenes ground truth; (**b**,**d**) BEVFusion results.

**Figure 9 sensors-24-02828-f009:**
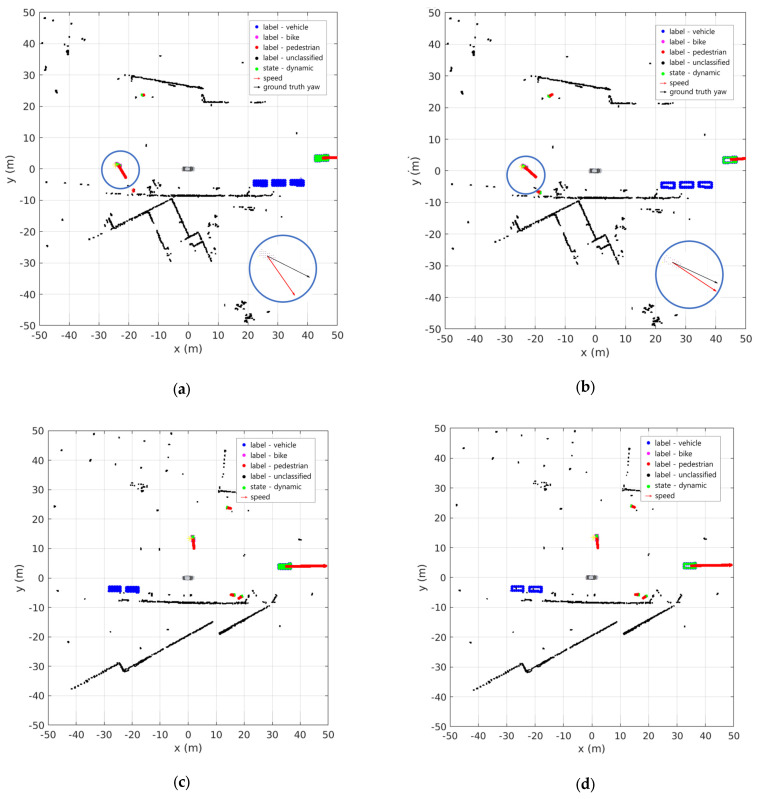
DOGM with semantic information results of scene no. 98 using (**a**,**c**) SBBUM and (**b**,**d**) EBBUM.

**Figure 10 sensors-24-02828-f010:**
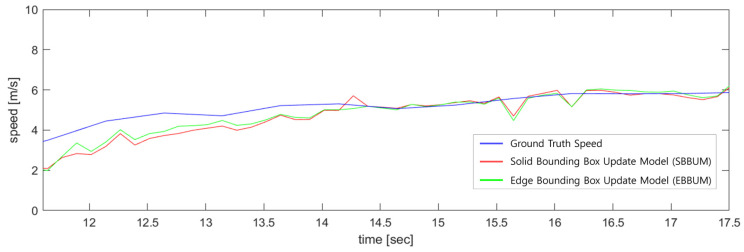
Comparison of speed estimations with ground truth using SBBUM and EBBUM. (scene no. 98).

**Figure 11 sensors-24-02828-f011:**
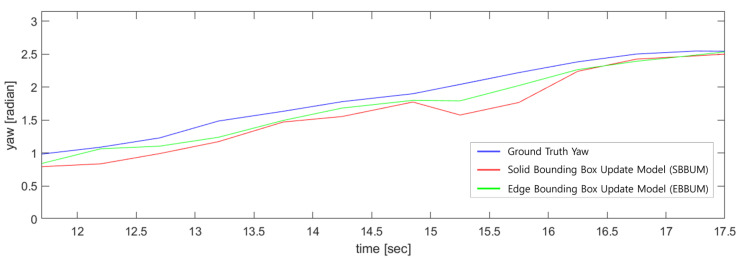
Comparison of direction estimations with ground truth using SBBUM and EBBUM. (scene no. 98).

**Figure 12 sensors-24-02828-f012:**
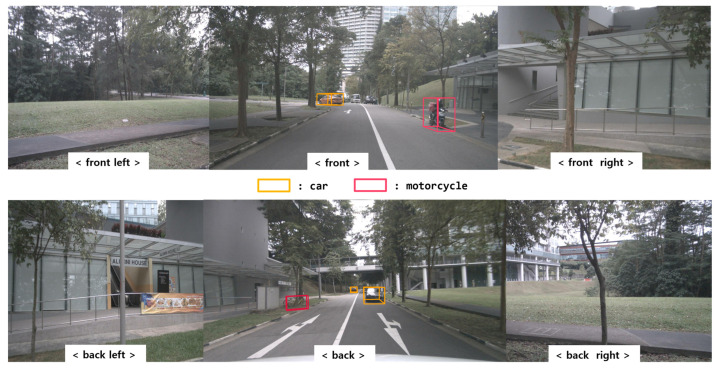
BEVFusion results with camera image (scene no. 272).

**Figure 13 sensors-24-02828-f013:**
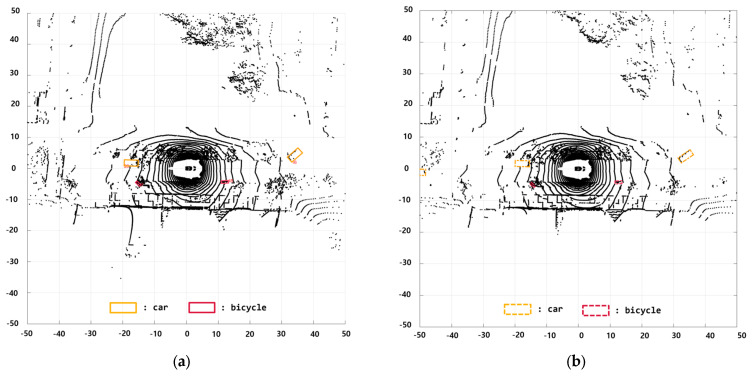
BEVFusion results with LiDAR measurements: (**a**,**c**) nuScenes ground truth and (**b**,**d**) BEVFusion.

**Figure 14 sensors-24-02828-f014:**
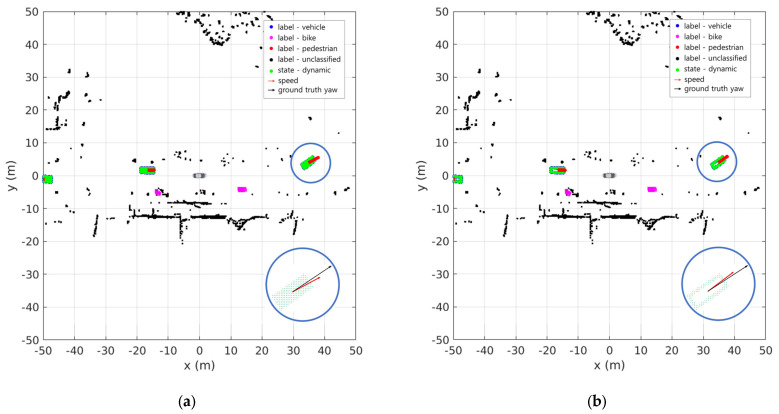
DOGMs with semantic information results using (**a**,**c**) SBBUM and (**b**,**d**) EBBUM.

**Figure 15 sensors-24-02828-f015:**
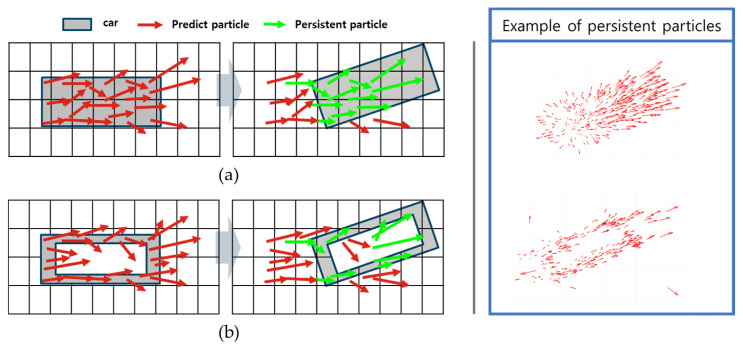
Examples of predicted and persistent particles using (**a**) SBBUM and (**b**) EBBUM.

**Figure 16 sensors-24-02828-f016:**
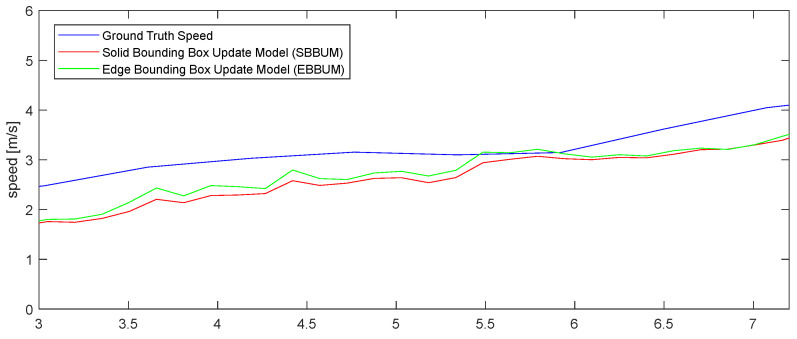
Comparison of speed estimations with ground truth using SBBUM and EBBUM. (scene no. 272).

**Figure 17 sensors-24-02828-f017:**
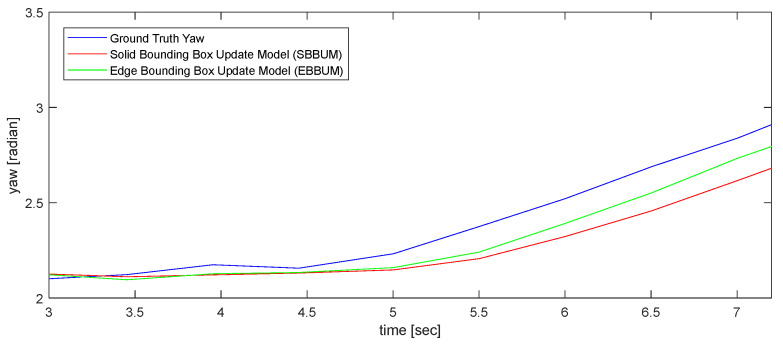
Comparison of direction estimations with ground truth using SBBUM and EBBUM. (scene no. 272).

**Table 1 sensors-24-02828-t001:** nuScenes sensor specifications.

Sensor	Specifications
LiDAR	Velodyne HDL-32E
Camera	Basler acA1600-60gc
RADAR	Continental ARS 408-21

## Data Availability

Data is contained within the article.

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
