# Peer review of "Dynamic Occupancy Grid Map with Semantic Information Using Deep Learning-Based BEVFusion Method with Camera and LiDAR Fusion"

_sensors, 2024, doi:10.3390/s24092828_

Round 1
Reviewer 1 Report
Comments and Suggestions for Authors
The paper presents a study on enhancing the detection and classification of objects in autonomous driving through a dynamic occupancy grid map (DOGM) that integrates deep learning-based camera and LiDAR sensor fusion. This approach overcomes the limitations of traditional DOGMs by enabling object classification and improving velocity estimation. The study introduces update models based on Dempster–Shafer evidence theory for accurate object and velocity classification and evaluates performance using the nuScenes dataset. However, the paper lacks proven contributions to solving the presented problems.
Comments and suggestions:
1 Based on the problem description in the paper, the authors attempt to solve dynamic occupancy grid mapping using deep learning object classification and state estimation methods. However, the majority of the results section focuses on showing the performance differences between SBBUM and EBBUM, leaving the paper's main goal unclear.
2 The title of the paper mentions deep learning-based camera and LiDAR detection. In the paper, the authors have no contributions to deep learning-based detection methods. Instead, they only applied BEVFusion. Therefore, the title should be updated to mention BEVFusion instead of deep learning-based camera and LiDAR detection. Additionally, since there is no contribution in the detection part, the authors should remove the related paragraphs in the introduction, related work review, and results sections.
3 In the results, the authors only compare the performance of using SBBUM and EBBUM. Many papers solve the DOGM problem. To highlight the contributions of this paper, the authors should compare the performance of their method with that of others in the field.
4 In Figure 18, the colors in the legend are incorrect.
5 In Section 3.2, there is a reference to equation 15, which does not exist in the paper.
Author Response
Thank you very much for taking the time to review this manuscript. Please find the detailed responses below and the corresponding corrections highlighted by red colors in the re-submitted files.

Reviewer 2 Report
Comments and Suggestions for Authors
1. Add the contents in the abstract of the paper and highlight the impact of the proposed work.
2. To explore Comparative results with existing approaches/methods relating to the proposed work.
3. The method/approach in the context of the proposed work should be written in detail.
4. In Section 3.2. Sensor Fusion DOGM, about Sensor Fusion, what data is fused in here?
5. In Section,athors have provided Evaluation Process, Evaluation Index?
6. What are the limitations behind this study? This topic should be highlighted somewhere in the text of manuscript
7. The inspiration of your work must be highlighted. I would suggest adding some recent literature in the manuscript. For instance:
Su H, Zhao D, Elmannai H, et al. Multilevel threshold image segmentation for COVID-19 chest radiography: A framework using horizontal and vertical multiverse optimization[J]. Computers in Biology and Medicine, 2022, 146: 105618.
Hussien A G, Heidari A A, Ye X, et al. Boosting whale optimization with evolution strategy and Gaussian random walks: An image segmentation method[J]. Engineering with Computers, 2023, 39(3): 1935-1979.
Wu S, Liu W, Wang Q, et al. Reffacenet: Reference-based face image generation from line art drawings[J]. Neurocomputing, 2022, 488: 154-167.
H. Zhao et al., "Intelligent Diagnosis Using Continuous Wavelet Transform and Gauss Convolutional Deep Belief Network," in IEEE Transactions on Reliability, vol. 72, no. 2, pp. 692-702, June 2023, doi: 10.1109/TR.2022.3180273.
and so on.
Comments on the Quality of English LanguageModerate editing of English language required
Author Response

(The authors gave the same response as above.)

Round 2
Reviewer 1 Report
Comments and Suggestions for Authors
The authors addressed all comments, even though the contributions of the paper are still limited. The authors may think about updating the contributions part in Section 1.
Author Response
The authors would like to thank the reviewer for taking precious time and effort to review this paper. As the reviewer mentioned, the authors modified the contribution part in Section 1 as follows.
"The robustness of environmental perception is enhanced by developing a DOGM that incorporates deep learning-based object recognition results through BEVFusion camera–LiDAR sensor fusion method."
Reviewer 2 Report
Comments and Suggestions for Authors
This paper can be accepted now.
Comments on the Quality of English LanguageMinor editing of English language required
Author Response
The authors would like to thank the reviewer for taking the previous time and effort to review this paper. The authors will do our best to improve the quality of the English language.